# Fast-Track Discovery of SARS-CoV-2-Neutralizing Antibodies from Human B Cells by Direct Functional Screening

**DOI:** 10.3390/v16030339

**Published:** 2024-02-22

**Authors:** Matthias Hillenbrand, Christoph Esslinger, Jemima Seidenberg, Marcel Weber, Andreas Zingg, Catherine Townsend, Barbara Eicher, Justina Rutkauskaite, Peggy Riese, Carlos A. Guzman, Karsten Fischer, Simone Schmitt

**Affiliations:** 1Memo Therapeutics AG, 8952 Schlieren, Switzerland; matthias.hillenbrand@memo-therapeutics.com (M.H.); marcel.weber@memo-therapeutics.com (M.W.); andreas.zingg@unibas.ch (A.Z.); barbara.eicher@memo-therapeutics.com (B.E.); justina.rutkauskaite@memo-therapeutics.com (J.R.); karsten_fischer@gmx.net (K.F.); simone.schmitt@memo-therapeutics.com (S.S.); 2Department of Vaccinology and Applied Microbiology, Helmholtz Centre for Infection Research, 38124 Braunschweig, Germany; peggy.riese@helmholtz-hzi.de (P.R.); carlosalberto.guzman@helmholtz-hzi.de (C.A.G.)

**Keywords:** SARS-CoV-2, neutralizing antibodies, antibody discovery, monoclonal antibodies, viral variants

## Abstract

As the COVID-19 pandemic revealed, rapid development of vaccines and therapeutic antibodies are crucial to guarantee a quick return to the status quo of society. In early 2020, we deployed our droplet microfluidic single-cell-based platform DROPZYLLA^®^ for the generation of cognate antibody repertoires of convalescent COVID-19 donors. Discovery of SARS-CoV-2-specific antibodies was performed upon display of antibodies on the surface of HEK293T cells by antigen-specific sorting using binding to the SARS-CoV-2 spike and absence of binding to huACE2 as the sort criteria. This efficiently yielded antibodies within 3–6 weeks, of which up to 100% were neutralizing. One of these, MTX-COVAB, displaying low picomolar neutralization IC50 of SARS-CoV-2 and with a neutralization potency on par with the Regeneron antibodies, was selected for GMP manufacturing and clinical development in June 2020. MTX-COVAB showed strong efficacy in vivo and neutralized all identified clinically relevant variants of SARS-CoV-2 at the time of its selection. MTX-COVAB completed GMP manufacturing by the end of 2020, but clinical development was stopped when the Omicron variant emerged, a variant that proved to be detrimental to all monoclonal antibodies already approved. The present study describes the capabilities of the DROPZYLLA^®^ platform to identify antibodies of high virus-neutralizing capacity rapidly and directly.

## 1. Introduction

When severe acute respiratory syndrome coronavirus 2 (SARS-CoV-2) caused a global pandemic in early 2020 [1], scientists around the globe embarked on the development of therapeutic antibodies [2,3,4,5,6,7,8,9,10] as a potential counter measure. While most of the approaches used convalescent, human donors [3,4,5,7,8,9,10], some approaches utilized transgenic mice [4,6] as a source of B cells. In most of the cases, the B cell population was first enriched for binding to the receptor-binding domain (RBD) of the SARS-CoV-2 spike protein, as this domain plays a pivotal role during infection by interaction with angiotensin-converting enzyme II (huACE2) or other potential receptors [1,11,12]. The methodologies used to run antibody discovery campaigns ranged from classical hybridoma technology [6] to more modern technologies including B cell cloning [4,5,10], single B cell sequencing [3,7,8] or microfluidic B cells assays [9,10]. We deployed our proprietary DROPZYLLA^®^ platform to join the race to find SARS-CoV-2-neutralizing antibodies in mid-March 2020. Our microfluidic antibody discovery technology used convalescent donors as a B cell source to clone and screen entire B cell repertoires. To that end, blood from 12 donors who had overcome PCR-test-proven coronavirus-induced disease 19 (COVID-19) with mild to moderate or moderate to severe symptoms was collected. Then, recombinant cognate antibody mammalian display libraries were generated by copying the antibody genetic information of single memory B cells from donors into HEK293T cells. The antibodies were expressed in HEK293T cells as membrane-bound full-length IgG (mIgG-HEK) to enable antigen-specific sorting. 

As the RBD of the SARS-CoV-2 spike protein interacts with human angiotensin-converting enzyme 2 (huACE-2) on the host cell for cell entry [13,14,15], we screened for cells that displayed antibodies that bound the RBD but blocked the binding of huACE2. Most antibodies that were identified from this screening proved to neutralize SARS-CoV-2 infection in subsequent assays. With this approach, it was possible to identify neutralizing antibodies within 3 to 6 weeks after receipt of the blood sample, allowing us to rapidly identify our development candidate MTX-COVAB and move it into the clinic.

During the course of the clinical development, new variants of SARS-CoV-2 evolved that acquired mutations in the spike protein, which not only influenced its affinity for huACE2 [12,16] but also its flexibility [17,18]. These changes affected the effectiveness of antibodies, due to changes in epitope or accessibility. The emergence of the Omicron variant not only halted the development of MTX-COVAB, but also proved to be detrimental to all other antibodies that were already used in the clinic [19,20,21]. Here, we report that one of our initial lead candidates was able to retain neutralization over a range of SARS-CoV-2 variants including Omicron, although neutralization potency was reduced for the latter variant.

## 2. Materials and Methods

### 2.1. Donors and Ethics

Criteria for blood donors to be included in our study were that they had recovered from COVID-19 and had experienced mild to moderate and moderate to severe symptoms. Blood draw (30 mL) was performed no earlier than two weeks after the last symptoms by a qualified medical doctor. On the day of the blood draw, peripheral blood mononuclear cells (PBMCs) were isolated by density gradient centrifugation using Sep-Mate™-50 tubes filled with Lymphoprep™ (both Stemcell Technologies, Vancouver, Canada). Memory B cells, defined to be the cell population positive for CD22 and negative for CD3, CD8, CD56, IgA, IgD, and IgM, were isolated by the BD FACS Aria III cell sorter. Donors were recruited in accordance with the laws of Switzerland and under ethics approval BASEC-2016-01260 of the Cantonal Ethics Commission of Zurich.

### 2.2. Expression, Purification and Fluorescent-Labelling of Coronavirus Spike RBD and huACE-2

The RBD of the SARS-CoV-2 spike protein (residues 319–541, Wuhan strain, UniProtKB: P0DTC2) with a C-terminal His tag and a soluble form of human ACE-2 (residues 1–740, UniProtKB: Q9BYF1) with a C-terminal rabbit Fc tag were transiently expressed in the HEK293F cells.

The expressed RBD was affinity purified from supernatants using HisPur™ Ni-NTA Spin columns (ThermoFisher, Waltham, MA, USA), whereas huACE2 was purified from supernatants using Protein G High Performance Spintrap™ columns (GE Healthcare, Chicago, IL, USA). The purity and integrity of proteins were checked by Coomassie-stained SDS-PAGE.

RBD and huACE-2 were labelled with fluorophores using the Lightning-Link^®^ Antibody Conjugation Kits (Abcam, Cambridge, UK).

### 2.3. Antibody Discovery

Memory B cells from peripheral blood were used to prepare cognate antibody repertoire expression libraries as described in patent WO/2015/121434. In short, memory B cells on a single cell level were lysed in microfluidic droplets in a volume of 300 pl. Upon lysis, the mRNA was captured on mRNA capture beads, and after RT-PCR, single beads were re-encapsulated in droplets and immunoglobulin heavy and light chain variable regions were amplified by droplet PCR using a set of primers [22,23] ensuring amplification of the full sequence repertoire. During this PCR reaction, cognate heavy and light chain pairing was conserved by linkage in a single amplicon. PCR fragments of linked immunoglobulin light chain and heavy chain variable regions were cloned into an expression cassette, providing the human immunoglobulin constant heavy region combined with a transmembrane domain derived from human CD8 to allow for mammalian cell display of the antibodies. HEK293T cells displaying membrane-bound full-length IgG (mIgG-HEK) were created by transduction of HEK293T cells with lentiviruses encoding the antibody repertoire at low MOI. Screening of the antibody libraries was performed using fluorescence-activated cell sorting (FACS) of antigen-specific cells using fluorescently labelled SARS-CoV-2 Spike RBD and huACE2. These sorts yielded RBD-specific-antibody-expressing HEK single cell clones. Clone specificity was confirmed in analytical flow cytometry, testing for high-affinity binding of cell-membrane-expressed antibodies to purified SARS-CoV-2 Spike RBD protein and, simultaneously, absence of binding to huACE2. The absence of huACE2 binding not only indicates a blocking of the specific interaction between RBD and huACE2, but also the absence of unspecific binding of the antibody to an unrelated antigen. Antibody clones specific for RBD binding and negative for huACE2 binding were sequenced.

### 2.4. Production of Human Monoclonal Antibodies

Antibody sequences of selected clones were PCR amplified and sub-cloned into the pCI expression vector for soluble IgG expression using the constant region derived from human IgG1. Sequence information of the variable region of antibodies REGN10933 and REGN10987 was obtained from [4] and were linked to the same constant region as our antibodies. Negative control/isotype control antibody (24C03) is a human-derived antibody specific for tetanus toxoid. The antibodies were expressed after transient transfection in 30–50 mL cultures of ExpiCHO cells following the standard protocol as set forward by the manufacturer (ThermoFisher, Waltham, MA, USA). The antibodies were then purified from cell supernatant using Protein G High Performance Spintrap™ columns (GE Healthcare, Chicago, IL, USA) as recommended by the manufacturer. The purity and integrity of the proteins were checked by Coomassie stained SDS-PAGE (reducing and non-reducing) and SEC-HPLC.

### 2.5. Preparation of Pseudo-Typed SARS-CoV-2 (SARS-CoPsV-2)

Lentivirus-based pseudo-typed virus particles with trimeric SARS-CoV-2 spike protein or a spike protein variant (Appendix A), replacing the VSV-G gene, were cloned and produced in-house similar to previous studies [2,6,24]. In order to increase surface expression, the SARS-CoV-2 spike protein was modified by a deletion of the furin cleavage site [25] (aa682-685, “RRAR”, replaced by a single alanine) and by a C-terminal truncation of the last 17 amino acids [26]. As a reporter of infection, a transfer vector encoding a constitutively expressed, secreted luciferase was co-transfected during production of pseudo-typed virus batches. Virus-containing supernatants were collected and treated with PEG-it™ virus precipitation solution (System Biosciences, Palo Alto, CA, USA) to concentrate SARS-CoV-2 pseudo-typed viral particles and remove secreted luciferase from the virus preparation. Pseudovirus particles in PBS were stored frozen at −80 °C as single-use aliquots.

### 2.6. Neutralization of SARS-CoPsV-2

HEK293T cells stably expressing full-length huACE2 (aa1-805; UniProtKB: Q9BYF1) were seeded and left to adhere for 4 h (15,000 cells/well). Serial antibody dilutions were prepared and added in triplicates to the adherent cells. Directly thereafter, SARS-CoPsV-2 carrying a secreted luciferase reporter gene was added, and the mix was incubated for 3 days. After this period, the amount of secreted luciferase in the supernatant was determined by luciferase assay (NanoGlo^®^ Luciferase Assay, #N1130, Promega, Madison, WI, USA) for the qualitative analysis of infected cells. The percentage of infectivity was calculated as the ratio of luciferase readout in the presence of neutralizing mAbs normalized to luciferase readout in the presence of an unrelated, non-neutralizing mAb (24C03; 100% infection) and the luciferase readout of pseudovirus alone diluted in growth medium (0% infection). The data were analyzed, and the half maximal inhibitory concentrations (IC50) were determined using the “log(Inhibitor) vs. normalized response—Variable slope” fitting model of GraphPad Prism 8 (GraphPad Software, Boston, MA, USA).

### 2.7. Binding to Variants of SARS-CoV-2 Spike

Variants of the SARS-CoV-2 spike protein were generated by gene synthesis. The SARS-CoV-2 spike protein contained the same modifications of the furin cleavage site and C-terminal truncation as described for pseudo-typed virus particles. The HEK293T cells were transfected with a SARS-CoV-2 spike protein-encoding plasmid using jetPRIME (Polyplus transfection, Illkirch, France). Two days after transfection, the SARS-CoV-2 spike protein expressing and wild-type HEK293T cells were gently detached using 4 mM EDTA in PBS. The cells were washed in wash buffer (PBS + 2 mM EDTA + 0.5% (*v*/*v*) FBS) and distributed to 96-well plates (~1 × 10^5^ cells/well) for staining. The antibodies to be tested for binding to spike protein were diluted to 20 nM (final concentration) in wash buffer and added to the cells. After 40 min at 4 °C, cells were washed, and the bound antibody was detected by a rat anti-huIgG-BV421 (1:200 dilution; Biolegend, San Diego, CA, USA). Alternatively, some wells were stained only with 8.1 nM of huACE2-AF647 to estimate the level of spike protein expression or effect of mutation on huACE2 binding. After another 40 min at 4 °C, cells were washed and analyzed by a flow cytometer. The fraction of cells bound to antibody or huACE2 was analyzed by FlowJo 10.7.1 (BD).

### 2.8. Neutralization of Wild-Type SARS-CoV-2

At day 0, Vero E6 cells were seeded with a concentration of 1 × 10^5^ cells/well in 300 µL medium. At day 1, the antibodies were serially diluted using a 1:3.16 dilution starting with 1 or 10 µg/mL in a total volume of 300 µL/well. Subsequently, the antibodies in triplicate were mixed with SARS-CoV-2 virus (1800 PFU/mL) and incubated for 1 h at 37 °C. Then, media from the Vero E6 cells were removed, and the cells were incubated with the antibody-virus mix for 1h. Afterwards, the inoculum was aspirated, and the cells were overlaid with 300 µL of 1.5% methylcellulose and incubated for 3 days. Then, the plates were fixed with 6% formaldehyde for 1 h followed by a 1 h staining with 300 µL crystal violet/well. After 2–3 days, the plaques were counted manually under an inverted light microscope.

### 2.9. In Vivo Experiments (Hamster)

A total of 56 Golden Syrian hamsters, 36 male and 20 female, weighing between 80 g and 130 g were used in the study. The animals were weighed prior to the start of the study and randomly distributed in the different cohorts. Each of the 5 prophylactic cohorts contained 4 male and 4 female hamsters. Each of the four therapeutic cohorts contained 4 male hamsters. The animals were monitored twice a day at least 6 h apart during the study period. Body weights were measured once a day during the study period. Antibodies were diluted in PBS and dosed at the indicated concentrations in a constant volume of 500 µL through intraperitoneal (IP) injection either one day before challenge with virus (“prophylaxis”) or one day after challenging with virus (“therapy”). Animals were challenged on day 0 with SARS-CoV-2 by administration of 0.05 mL of a 1:10 dilution of SARS-CoV2 (OWS stock, CAT#-NR-53780), into each nostril.

### 2.10. Antibody-Dependent Cellular Cytotoxicity (ADCC)

Target cell preparation. Target cells (HEK293T wt cells, HEK293T cells stably expressing SARS-CoV-2 spike protein, or Raji cells) at 1 × 10^7^ cells/mL were labelled with 2 µM PKH67 green fluorescent cell linker mini kit (Sigma-Aldrich, St. Louis, MO, USA) for general membrane labelling according to the protocol provided by the manufacturer. Labelling was checked by FACS analysis for homogeneous staining well above background (~100-fold). Transfer of the dye to unlabeled cells after overnight co-culture was minimal, and target cell population is clearly identifiable.

NK cell isolation. Primary human NK cells were isolated from cryopreserved PBMCs using the human NK Cell Isolation Kit (Miltenyi, Bergisch Gladbach, Germany) according to provided protocol. Based on the analysis by flow cytometry for CD56^+^ CD3^−^ cells, the isolated NK cell populations used for the assay were >80% pure. PBMCs were obtained from buffy coat.

ADCC assay. Labelled target cells in RPMI-1640 + 10% (*v*/*v*) FBS, 2 mM L-glutamine, 1 mM sodium pyruvate, 10 mM HEPES, and 4.5 g/L D-(+)-Glucose were seeded in triplicates at 5000 cells/well in 96-well plates and left to adhere (HEK293T) or sit (Raji) for 2–3 h. After the addition of 4-fold serially diluted antibody (8-point dilution ranging from 50 nM to 3 pM final concentration), primary human NK cells (25,000 cells/well) were added to target cells. As non-specific background control, wells only containing target and effector cells were set up as well. After overnight incubation at 37 °C, 5% (*v*/*v*) CO_2_, cells were stained with SYTOX™ Blue Dead Cell Stain (ThermoFisher, Waltham, MA, USA), and percentage of dead target (PKH67-positive) cells was determined by flow cytometry (CytoFlex S, BeckmanCoulter, Brea, CA, USA). Obtained data were analyzed by FlowJo 10.7.1 (Becton Dickinson & Company, Franklin Lakes, NJ, USA). Data were normalized using the non-specific background control (no antibody; 0% ADCC) and plotted using GraphPad Prism 8 (GraphPad Software, Boston, MA, USA).

### 2.11. Antibody-Dependent Cellular Phagocytosis (ADCP)

Target cell preparation. Target cells (HEK293T wt cells, HEK293T cells stably expressing SARS-CoV-2 spike protein, or Raji cells) were labelled with PKH67 green fluorescent cell linker mini kit (Sigma-Aldrich, St. Louis, MO, USA), as described for ADCC.

Effector cell preparation. Frozen PBMCs were thawed, resuspended at 1 × 10^6^ cells/mL in PBS, and stained with 5 µM CellTrace™ Violet dye (ThermoFisher, Waltham, MA, USA) for 20 min at 37 °C. Excessive dye was quenched by addition of 10% (*v*/*v*) FBS. PBMCs were obtained from buffy coat.

ADCP assay. Labelled target cells in RPMI-1640 + 10% (*v*/*v*) FBS, 2 mM L-glutamine, 1 mM sodium pyruvate, 10 mM HEPES, and 4.5 g/L D-(+)-Glucose were seeded in triplicates at 7500 cells/well in 96-well plates and left to adhere (HEK293T) or sit (Raji) for 2–3 h. After the addition of 4-fold serially diluted antibody, labelled PBMCs (150,000 cells/well) were added to target cells. After overnight incubation at 37 °C, 5% (*v*/*v*) CO_2_, cells were stained with anti-human CD14-APC (1:50). After washing, cells were fixed with 4% paraformaldehyde in PBS for 10 min and analyzed by flow cytometry (CytoFlex S, BeckmanCoulter, Brea, CA, USA). The proportion of PKH67-positive, CD14+/CellTrace Violet+ monocytes was determined by analysis in FlowJo 10.7.1 (Becton Dickinson & Company, Franklin Lakes, NJ, USA). The background level was normalized using wells containing all components of the assay except the antibodies, and the data were plotted using GraphPad Prism 8 (GraphPad Software, Boston, MA, USA).

### 2.12. Complement-Dependent Cytotoxicity (CDC)

Human serum preparation. Native blood of 10 human donors was pooled and centrifuged at 1900 g for 10 min. The resulting supernatant was transferred to a fresh tube and spun again at 16,000 g. The final supernatants were aliquoted and kept frozen at −80 °C.

CDC assay. HEK293T wt cells, HEK293T cells stably expressing SARS-CoV-2 spike protein, or Raji cells were seeded in triplicates at 25,000 cells/well in their respective medium (25 µL/well). HEK293T cells were allowed to adhere for 2 h in the incubator. Antibodies were serially diluted 4-fold (8-point dilution ranging from 667 nM to 41 pM final concentration) in HEK293T or Raji medium and added to the cells in triplicate (25 µL/well). After 10 min of incubation, 25 µL of prediluted human serum was added at a final concentration of 20% (*v*/*v*). To determine the non-specific background, wells containing all components of the assay except the antibodies were set up. After overnight incubation at 37 °C, 5% (*v*/*v*) CO_2_, cells were stained with SYTOX™ Blue Dead Cell Stain (ThermoFisher, Waltham, MA, USA), and the percentage of dead cells was determined by flow cytometry (CytoFlex S, BeckmanCoulter, Brea, CA, USA). To obtain the maximal signal of dead cells, the cells were heated at 65 °C for 15 min. The obtained data were analyzed by FlowJo 10.7.1 (Becton Dickinson & Company, Franklin Lakes, NJ, USA). The data were normalized using the non-specific background and dead cell controls and plotted using GraphPad Prism 8 (GraphPad Software, Boston, MA, USA).

## 3. Results

### 3.1. Direct Discovery of SARS-CoV-2-Neutralizing Antibodies

Screening for virus-neutralizing antibodies was performed using our DROPZYLLA^®^ platform. After drawing blood from 12 convalescent COVID-19 patients that had experienced mild to moderate or moderate to severe symptoms, we copied the antibody genetic information of total ~2 million single memory B cells into HEK293T cells. The antibodies, maintaining cognate heavy and light chain pairing of the original B cell, were expressed as membrane-bound full-length IgG (Figure 1A).

The generated libraries, copies of their original memory B cell repertoires, were sorted individually on a fluorescence-activated cell sorter (Figure 1B): In the first round, we enriched clones that express an antibody binding to RBD. For this sort, we used a dual labelling approach by utilizing RBD labelled either with PE or APC and only double-positive clones were sorted. In the second round, we introduced a functional read-out by staining cells with RBD-PE and huACE2-Alexa Fluor 647 (AF647), as our desired mode of action was blocking virus attachment to the cell. Therefore, only cells that efficiently blocked binding of RBD to huACE2, i.e., fell into gate Q4 (Figure 1B; RBD-PE^+^, huACE2-AF647^−^), were single-cell-sorted. Sorted clones were once more confirmed by analytical flow cytometry for RBD-binding and absence of huACE2 binding (Appendix A). The hits of this screen were sequenced, subcloned into soluble IgG format, expressed, and purified for further characterization.

### 3.2. Characterization of Antibodies in Neutralization of SARS-CoPsV-2 and Clinical Isolate

All antibody hits coming out of our screen were tested for virus neutralization using lentivirus-based virus particles pseudo-typed with trimeric SARS-CoV-2 spike. Based on this assay and the resulting IC50 values, we ranked our candidate antibodies and nominated four lead candidates termed CoVAb 5, 13, 36, and 47 (Figure 2A and Table 1). At the time of antibody discovery, the only reported variant besides the original Wuhan strain was the D614G mutation in the spike protein [27]. All four lead candidates also neutralized this variant (Figure 2B) with only minor changes in potency (Table 1). Encouraged by the results on both virus variants, we tested the lead candidates in a plaque assay with a replication competent SARS-CoV-2 clinical isolate to confirm neutralization potency. Indeed, neutralization potency and ranking based on pseudovirus neutralization assays could be confirmed on the clinical isolate (Figure 2C and Table 1). Interestingly, three of the antibodies were raised from donors that had experienced only mild symptoms, while one was raised from a donor that had severe symptoms and was treated in the intensive care unit (ICU, Figure 2D). Due to the urgency of the situation, the best performing antibody in both assays, CoVAb 36, was selected as the development candidate, since it combined excellent virus neutralization activity with favorable developability attributes (Appendix A). GMP manufacturing of CoVAb 36, under the name MTX-COVAB, was initiated at the beginning of August 2020, five months after the initiation of the project. While GMP manufacturing was already ongoing, further assays to characterize MTX-COVAB were conducted.

### 3.3. MTX-COVAB Neutralizes Wild-Type SARS-CoV-2 Efficiently in an In Vivo Hamster Model

To show in vivo efficiency of MTX-COVAB, the Syrian golden hamster model for SARS-CoV-2 infection was chosen. This model has increased relevance to the human disease in terms of strong lung pathology and a dramatic loss in body weight, which was in turn our main read-out for efficacy of the antibody [2,28,29]. We assessed MTX-COVAB both as a prophylaxis and a therapy (Figure 3). When used as a prophylaxis, MTX-COVAB prevented weight loss even at day 1 after virus inoculation. With all doses tested, including the lowest dose administered (1 mg/kg), a gain of weight was visible in a dose-dependent manner that reached a plateau at 10 mg/kg body weight (Figure 3A). When used as a therapy, a dose-dependent recovery from weight loss could be observed. This therapeutic effect was most striking on day 7, where a maximal weight loss of 15% was observed with placebo, compared to less than 3% with the two higher doses of 10 and 50 mg/kg. In summary, we demonstrated that MTX-COVAB is able to reduce the clinical symptoms of infection both in a therapeutic and prophylactic setting.

### 3.4. Fc-Effector Functions

Fc-mediated effector functions such as antibody-dependent cellular cytotoxicity, antibody-dependent cellular phagocytosis, and complement-dependent cytotoxicity (ADCC, ADCP, and CDC) could be important components of the anti-infectious activity of SARS-CoV-2-neutralizing antibodies.

For this reason, stable SARS-CoV-2-spike-protein-transduced HEK293T cells were generated as targets. To assay MTX-COVAB for ADCC, purified human NK cells were used as effector cells, and antibody-dependent killing of target cells was assessed. In this assay, MTX-COVAB showed a strong, dose-dependent, ADCC effect (Figure 4A).

When testing for ADCP using whole human PBMC as effector cells and measuring the uptake of target cells into cells of the monocyte lineage, MTX-COVAB again showed a dose-dependent effect (Figure 4B).

Lastly, CDC as the antibody-dependent killing of target cells in the presence of human serum was examined. MTX-COVAB showed a less pronounced, but still measurable effect (Figure 4C).

All three Fc-mediated effector functions were found to be specific for HEK293T cells expressing spike protein, and MTX-COVAB showed no effect on HEK293T wt and Raji cells. Taken together, MTX-COVAB mediates strong Fc-receptor-dependent cytotoxicity, which may well constitute an additional mechanism of action against the virus.

### 3.5. Comparison to Other Neutralizing Antibodies

In an attempt to benchmark the virus-neutralizing activity of MTX-COVAB to one of its peers, we compared it side-by-side with the two antibodies that form Regeneron’s cocktail [4] using our pseudovirus assay. MTX-COVAB shows a neutralization capacity that is on par with the more potent antibody of the cocktail (REGN10933) as well as with the cocktail (Figure 5).

We also submitted our antibody to the Coronavirus Immunotherapeutic Consortium (CoVIC) [30], where it is listed under CoVIC ID 300. Within the over 400 submitted antibodies, our antibody was within the best 10% antibodies regarding the IC50 in pseudovirus neutralization (Wuhan strain), highlighting again the capability of our technology to find potent antibodies.

### 3.6. Spike Mutant Binding and Neutralization

In the initial phase of the pandemic, when it was still unclear if and how the virus would evolve, researchers performed in vitro selections [2,31] or modelling [32] to identify potential future escape variants of the SARS-CoV-2 spike protein. To check if MTX-COVAB would still bind those reported, hypothetical, or rare mutations, we initially performed binding experiments. Thus, we generated HEK293T cells expressing the SARS-CoV-2 spike protein harboring selected mutations in the RBD and assessed the binding of MTX-COVAB in comparison to the wild-type SARS-CoV-2 spike protein. In addition, as a surrogate of the ability of that mutant to infect cells, we tested the binding of huACE2 as well. All the tested mutants contain only a single mutation compared to the original Wuhan strain (Figure 6).

MTX-COVAB, when compared to the binding of huACE2, bound most of the RBD variants. Mutations, including V367F, N439K, V445L, G446V, N501Y, and V503F, which had an impact on binding of MTX-COVAB also influenced binding of huACE2, which may be because of overall lower transfection or surface expression efficiency of that mutant or the impact of that mutation on binding of both MTX.COVAB and huACE2. The latter would indicate that the mutation might also have an impact on the ability to infect cells, since attachment to huACE2 is a prerequisite for a successful infection. Three mutations, E484K, Q493K, and S494P, affected binding of MTX-COVAB, but not huACE2, indicating that those mutations could potentially be escape variants. All those mutations had been found in in vitro selections [2,31], and the binding assay is only an approximation for the effect on virus infection and neutralization. Therefore, those findings had no direct consequence on the further development of MTX-COVAB.

Due to the limited informative value of the binding assay, we tested the effect of mutations in SARS-CoV-2-pseudovirus neutralization assays. When a new SARS-CoV-2 variant (Alpha or B.1.1.7) was identified in the UK around December 2020 [33], we could show that MTX-COVAB was able to fully neutralize this variant, but lost 10-fold in potency (Figure 7 and Table 2). We continued to monitor and check every new SARS-CoV-2 variant that emerged during the pandemic for effect on neutralization potency and efficacy. While the Alpha (B.1.1.7) variant was widely spread globally at the beginning of 2021 [34], other variants like Beta (B.1.351) and Gamma (B.1.1.248) were also reported but were more locally restricted to Africa and South America. Although both, Beta and Gamma, were able to escape neutralization by MTX-COVAB, the next global variant Delta (B.1.617.2) and its sub-variant Delta plus (AY.1) could be neutralized by MTX-COVAB with similar potency as the Alpha variant (Figure 7 and Table 2). Unfortunately, MTX-COVAB was unable to neutralize the next global variant Omicron (B.1.1.529), which ultimately brought the program to a stop.

Interestingly, one of our original lead candidates, CoVAb 47, showed neutralization of all variants tested, including Omicron (Figure 7). Although its neutralization potency dropped to ~100 nM (1000-fold) on the Omicron variant, it could neutralize all other variants with sub-nM IC50s, which is remarkable for an antibody obtained from a patient in the early phase of the pandemic.

## 4. Discussion

In the setting of a pandemic, rapid development and testing of new medicines is critical. Therefore, a drug discovery platform is needed that can deliver those new drugs within a short period of time. In addition, the usually long timelines of CMC and GMP manufacturing, clinical testing, and regulatory processes must be shortened, e.g., by parallelization of processes [35,36].

Here, we have shown that our DROPZYLLA^®^ platform is well suited for antibody discovery in a rapid pandemic response setting. Human antibodies are well suited to treat infectious diseases [37] and represent a drug class that is well characterized with a usually good PK and safety profile and low drug–drug interactions. Those characteristics help to significantly speed up timelines from bench to bedside in a global emergency, although time from discovery to bedside is still in the range of 10 months [19].

Although our platform infrastructure was not prepared for a rapid pandemic response, we managed to identify a potent neutralizing antibody, MTX-COVAB, within 2.5 months. During the first month, we obtained ethical approval, launched a campaign to recruit blood donors (>200 volunteers), and organized blood collection (12 donors). We were able to develop the neutralization assay and further streamline our workflow, including the development of a direct functional selection step.

At the time of its nomination, MTX-COVAB potently neutralized all known SARS-CoV-2 variants in vitro and showed potent in vivo activity against SARS-CoV-2 infection in the Syrian hamster model. Among all antibodies identified by the research community, MTX-COVAB was within the group of the most potent antibodies, as shown by comparison to all antibodies submitted to the CoVIC consortium. MTX-COVAB is an unmodified human IgG1 antibody that retains the ability to mediate ADCC, ADCP, and CDC, which may play an important role in therapeutic protection [38].

As is the nature of viruses, they tend to mutate and form new variants. This evolutionary process is driven by adaptation of the virus to its host in terms of infectivity, immune response, vaccination, and drug intervention. The only limitation in this evolutionary process is that the virus still needs to be able to infect the host, which involves its interaction with the host receptor(s). For that reason, we had chosen to block RBD/huACE2 interaction as this interaction involves conserved residues, which makes it harder for the virus to escape. Another strategy to limit viral escape is the use of two antibodies against different epitopes. This strategy had been followed by some pharma companies, for example, REGEN-COV (casirivimab + imdevimab) by Regeneron, bamlanivimab + etesevimab by Lilly, and Evusheld (tixagevimab + cilgavimab) by AstraZeneca. Although the strategy of using two antibodies may be feasible for larger corporations regarding CMC efforts and costs, this strategy was not feasible for a company of our size. Due to the urgent unmet medical need and despite the risk and uncertainty of how the virus would evolve, we decided to GMP-manufacture MTX-COVAB and test it in the clinic. While this was underway, we closely monitored newly emerging SARS-CoV-2 variants and could confirm that MTX-COVAB was able to neutralize globally predominant variants like Alpha (B.1.1.7) and Delta (B.1.617.2). While we were happy to see those results, we found MTX-COVAB to fail to neutralize other more regionally restricted variants (Table 2). With the advent of the Omicron (B.1.1.529) variant, the next globally predominant variant after Delta, we decided to halt our efforts, as MTX-COVAB was unable to neutralize this variant. Intriguingly, antibody cocktail approaches also ultimately failed to neutralize the new Omicron variant, and emergency use authorization was discontinued [19,20,21].

While we were testing emerging variants, we also tested our other lead candidates for neutralization. One of our leads, CoVAb 47, was constantly standing out, as it could neutralize all tested variants with relative low impact on neutralization potency, except Omicron (Table 2). Omicron affected neutralization potency significantly (IC50 ~100 nM), but the antibody was still able to neutralize this variant. This finding is quite impressive and may be because this antibody originated from a donor with severe symptoms that had been treated in ICU.

Over the last years, we have experienced an emergence of powerful computational approaches [39,40,41,42], which would allow us to check lead antibodies against future pandemic viruses for their ability to bind potential variants or to identify promising epitopes on targets. Although the decision process for clinical development would likely be delayed initially, as input data may not be available, the inclusion of in silico generated data, next to classic neutralization potency, affinity, and developability, may result in longer lasting clinical efficacy. Another strategy against viral escape by new strains is to develop pan-variant reactive antibodies, although their mode of action is not necessarily neutralization [43], or several infections or vaccinations of an individual are needed to develop such an immune response [44,45] or the epitopes to target. Nevertheless, in an urgent pandemic setting, the question remains if such strategies would be economically viable compared to the obvious strategy of isolating neutralizing antibodies from survivors of the initial pandemic phase. Overall, we were able to show that the careful selection of donor material in combination with our powerful DROPZYLLA^®^ platform can rapidly deliver rare and potent antibodies, which opens the avenue to a quick response to an emerging pandemic.

## Figures and Tables

**Figure 1 viruses-16-00339-f001:**
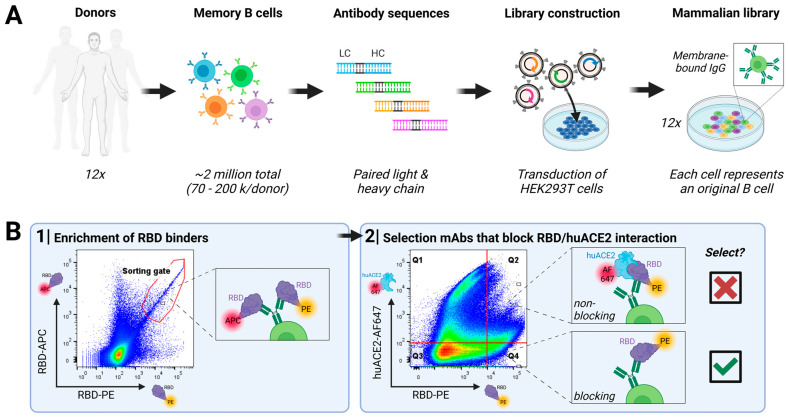
Discovery of SARS-CoV-2-neutralizing antibodies targeting RBD from convalescent human donors: (**A**) Workflow employed: Peripheral blood from convalescent donors was collected, and memory B cells were isolated. Cognate antibody repertoire libraries of each donor were generated using the DROPZYLLA^®^ platform. Antibodies were displayed on the cell surface of HEK 293T cells, suitable for FACS sorting. In every step, it was ensured that input memory B cell count was oversampled. (**B**) Two rounds of FACS sorting were employed to identify antibody clones positive for RBD binding and negative for huACE2 binding. The first round enriched the libraries for RBD binders using a dual-labelling strategy (RBD labelled with either APC or PE), resulting in between 20,000 to 40,000 cells enriched for each library, which were regrown for three days. In the second round, RBD binders that block the interaction with huACE2 (gate Q4) were single-cell-sorted (~500 clones/library), confirmed once more by analytical flow cytometry (Appendix A), and the respective antibody sequence was then retrieved. Unique sequences (>100 total) were sub-cloned into expression vectors for soluble IgG expression and characterization in downstream assays. Libraries screened with this strategy resulted in up to 100% of the antibodies that showed neutralizing activity on cell-based assays.

**Figure 2 viruses-16-00339-f002:**
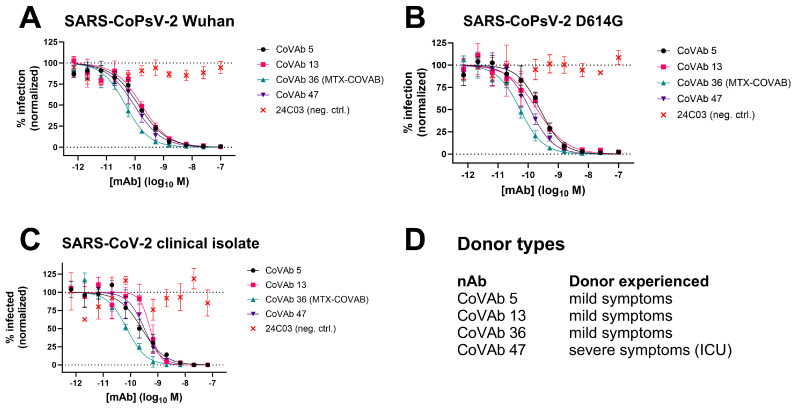
Neutralization data of four lead candidate antibodies CoVAb 5, 13, 36, and 47 and their donor type. As isotype/negative control 24C03 was used. Data are presented as normalized % infection at given final concentration of the antibody. Data shown are mean of triplicates, and error bars show SD. (**A**) Neutralization of pseudo-typed SARS-CoV-2 (SARS-CoPsV-2) particles with spike protein of the Wuhan strain. Serially diluted antibody was added to HEK293T cells expressing huACE2, followed by SARS-CoPsV-2 carrying a secreted luciferase reporter gene. After three days, the amount of luciferase in the supernatant was quantified as a read-out of infection. (**B**) as A, but with SARS-CoPsV-2 particles with spike protein of the D614G mutant. (**C**) Neutralization of replication-competent SARS-CoV-2 clinical isolate. Serially diluted antibody was incubated with SARS-CoV-2 and added to Vero E6 cells. After three days, the number of plaques were determined as a read-out of infection. (**D**) Indication if the donor, from which the antibody was isolated, experienced mild symptoms or severe symptoms and had to be treated in an intensive care unit (ICU).

**Figure 3 viruses-16-00339-f003:**
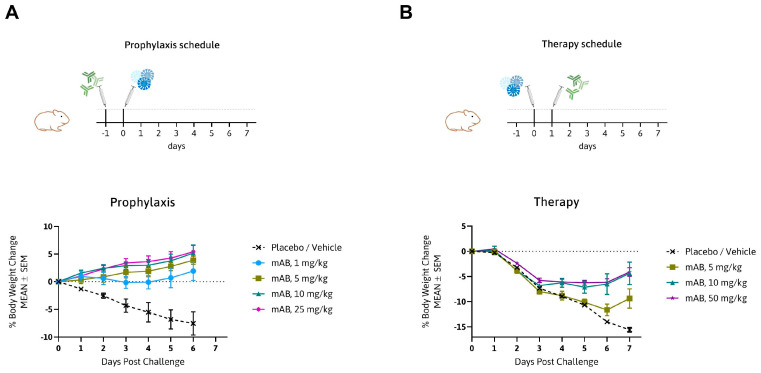
In vivo efficacy of MTX-COVAB for prophylaxis and therapy of SARS-CoV-2 infection. (**A**) Top. Outline of prophylactic study in the Syrian hamster model: animals were treated with MTX-COVAB dosed at 1 mg/kg, 5 mg/kg, 10 mg/kg, or 25 mg/kg one day before challenge with virus. Bottom. Time course of the % change in body weight. (**B**) Top. Outline of therapeutic study in the Syrian hamster model: animals were treated with MTX-COVAB dosed at 5 mg/kg, 10 mg/kg, or 50 mg/kg one day after challenge with virus. Bottom. Time course of the % change in body weight.

**Figure 4 viruses-16-00339-f004:**
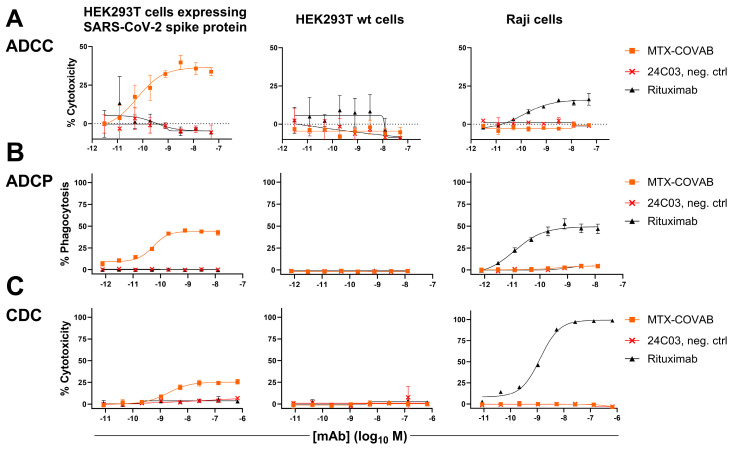
Anti-SARS-CoV2-Spike antibody MTX-COVAB induces specific ADCC, ADCP, and CDC. HEK293T cells expressing SARS-CoV-2 Spike protein, parental HEK293T cells, or Raji (CD20+) cells were mixed with (**A**) NK cells for ADCC, (**B**) with PBMCs for ADCP, and (**C**) with 20% human serum for CDC. A titration of anti-SARS-CoV2-Spike protein antibody MTX-COVAB, the IgG1 isotype control 24C03 or anti-CD20 antibody Rituximab was added to cells and incubated overnight. Cytotoxicity on target cells (**A**,**C**) was assessed using SYTOX™ Blue Dead Cell Stain, whereas phagocytosis (**B**) was assessed by the uptake of target cells by CD14+ monocytes. The baseline was defined as the average percent cytotoxicity/phagocytosis of the target/effector cell co-culture in absence of antibody. Each condition was tested in triplicate. Like Rituximab, which only affected Raji cells, MTX-COVAB only showed ADCC, ADCP, and CDC on SARS-CoV-2 expressing target cells, but not on HEK293T wt and Raji cells. Data shown are mean of triplicates, and error bars show SD.

**Figure 5 viruses-16-00339-f005:**
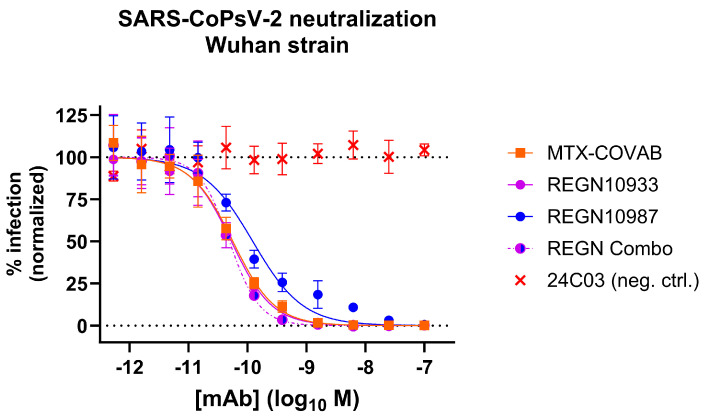
Comparison of MTX-COVAB with REGN-CoV-2 antibodies and cocktail (combo; REGN10933 and REGN10987 in a mixture of equal concentrations) in SARS-CoV-2-pseudovirus (SARS-CoPsV-2) neutralization assay. Indicated are percent infection of HEK293T-huACE2 cells. While 100% reflects the average signal from isotype control antibody (negative control, mAb 24C03) and 0% reflects the background signal. Data shown are mean of triplicates, and error bars show SD.

**Figure 6 viruses-16-00339-f006:**
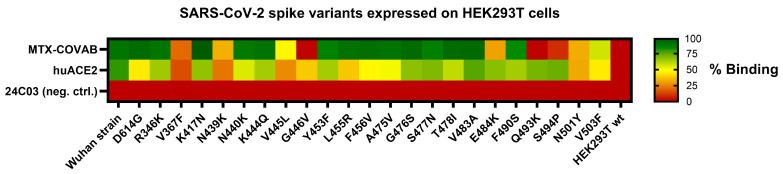
Binding of MTX-COVAB to single-point mutations in SARS-CoV-2 spike protein transiently expressed on HEK293T cells. Cells were incubated with MTX-COVAB, an irrelevant control antibody (24C03) and the cellular receptor of SARS-CoV-2, huACE2. Relative binding strength is indicated by colors where dark green represents strong binding and dark red no binding.

**Figure 7 viruses-16-00339-f007:**
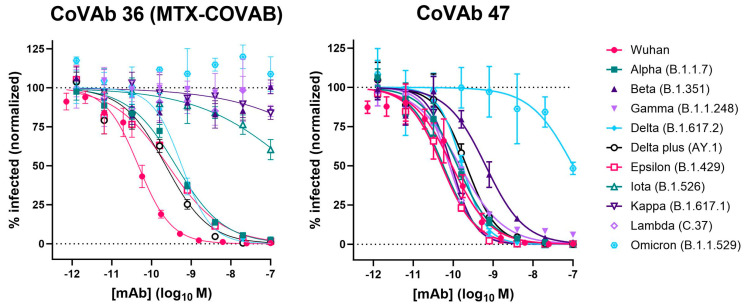
Neutralization of various virus variants by MTX-COVAB and CoVAb 47 in SARS-CoV-2-pseudovirus (SARS-CoPsV-2) assays. Neutralization of pseudo-typed SARS-CoV-2 (SARS-CoPsV-2) particles with spike protein of the indicated variant. Serially diluted antibody was added to HEK293T cells expressing huACE2 followed by SARS-CoPsV-2 carrying a secreted luciferase reporter gene. After three days, the amount of luciferase in the supernatant was quantified as a read-out of infection. Data are presented as normalized % infection at given final concentration of the antibody. Data shown are mean of triplicates, and error bars show SD.

**Table 1 viruses-16-00339-t001:** Lead candidate antibodies CoVAb 5, 13, 36, and 47 efficiently neutralize pseudo-typed SARS-CoV-2 (SARS-CoPsV-2) of Wuhan strain and D614G mutant, as well as replication-competent SARS-CoV-2 clinical isolate. CoVAb 36 was nominated as lead antibody and renamed to MTX-COVAB. IC50 values in nM are shown, and the corresponding 95% confidence interval (CI) as calculated by Prism is given.

	SARS-CoPsV-2 Wuhan		SARS-CoPsV-2 D614G		SARS-CoV-2 Clinical Isolate	
	IC50 (nM)	95% CI of IC50 (nM)	IC50 (nM)	95% CI of IC50 (nM)	IC50 (nM)	95% CI of IC50 (nM)
CoVAb 5	0.134	0.108 to 0.165	0.250	0.199 to 0.313	0.260	0.193 to 0.350
CoVAb 13	0.170	0.126 to 0.227	0.199	0.140 to 0.282	0.523	0.399 to 0.667
CoVAb 36 (MTX-COVAB)	0.047	0.039 to 0.056	0.054	0.045 to 0.064	0.074	0.060 to 0.092
CoVAb 47	0.093	0.072 to 0.119	0.117	0.093 to 0.147	0.321	0.237 to 0.433

**Table 2 viruses-16-00339-t002:** Neutralization potency of MTX-COVAB and CoVAb 47 on various virus variants (sequence alignment in Appendix A). IC50 values in nM are shown, and the corresponding 95% confidence interval (CI) as calculated by Prism is given. ND: no neutralization detected.

	MTX-COVAB		CoVAb 47	
	IC50 (nM)	95% CI of IC50 (nM)	IC50 (nM)	95% CI of IC50 (nM)
Wuhan	0.047	0.039 to 0.056	0.093	0.072 to 0.119
Alpha (B.1.1.7)	0.415	0.340 to 0.506	0.129	0.104 to 0.159
Beta (B.1.351)	ND	ND	0.652	0.478 to 0.888
Gamma (B.1.1.248)	ND	ND	0.148	0.124 to 0.177
Delta (B.1.617.2)	0.581	0.461 to 0.736	0.169	0.126 to 0.226
Delta plus (AY.1)	0.229	0.161 to 0.322	0.214	0.165 to 0.278
Epsilon (B.1.429)	0.261	0.188 to 0.361	0.052	0.043 to 0.063
Iota (B.1.526)	~400	133.8 to 2319	0.057	0.051 to 0.063
Kappa (B.1.617.1)	ND	ND	0.091	0.079 to 0.105
Lambda (C.37)	ND	ND	0.081	0.068 to 0.096
Omicron (B.1.1.529)	ND	ND	~100	56.15 to 387.4

## Data Availability

The data presented in this study are available on request from the corresponding author.

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
