# Peer review of "Fast-Track Discovery of SARS-CoV-2-Neutralizing Antibodies from Human B Cells by Direct Functional Screening"

_viruses, 2024, doi:10.3390/v16030339_

Round 1

Reviewer 1 Report

Comments and Suggestions for Authors

The study by Hillenbrand et. Al. reports an antibody discovery platform (DROPZYLLA) that was utilized towards the discovery of antibodies against SARS-CoV2 that can be used for therapeutic and prophylactic purposes. The theme of the article is based on the fact that in a pandemic scenario there is a rapid need to develop and discover antibodies with high potency that can be used as therapeutics against the infectious agent. The authors report the discovery of 4 antibodies of which one (MTX-COVAB) proceeded for in-depth functional characterization. MTX-COVAB was comparable to Regeneron antibodies with neutralization potency in picomolar range. Using the DROPZYLLA platform the authors claim that new therapeutic antibodies can be identified within 3-6 weeks.

The introduction section can be improved to describe the other methodologies that are available for antibody discovery. Also should include the limitations that the antibody discovery platforms will depend on the availability of the individuals that have recovered and will still be well within a 6 month mark until the antibody will be available for any medical use.

The platform and the methodology to select the antibodies is interesting and unique. The method section is well described and the results are well presented. However, including the designated steps that comprise the DROPZYLLA platform would be helpful (Fig 1). Also, PCR details for extracting the nucleic acid info from the memory B cells should be presented as it is this step that gives the technology an edge compared to other available methods.

Two different versions of neutralization assays are used for which the rationale is not provided. Plaque forming units assay could have been conducted with both the pseudovirus and the wildtype SARS-CoV2. Why was there no standard 1H incubation step with the Ab and the virus in luciferase based neutralization assay?

The authors have published same results in the bioRxiv as pre-print. The Fig showing the binding of MTX-COVAB (Fig 6 in this manuscript and Fig 2 in bioRxiv), has some discrepancies even though the scale used is comparable. The authors should verify what is the difference between the two that is responsible for the difference in the presented data.

Reviewer 2 Report

Comments and Suggestions for Authors

Please, see the attached PDF.

Reviewer 3 Report

Comments and Suggestions for Authors

In this paper, several clones of human mAbs were generated that effectively neutralized SARS-CoV-2. The authors have done a very voluminous work. The resulting antibodies were characterized by a wide range of methods. In general, the manuscript makes a very favorable impression.

To date, a fairly large number of human neutralizing antibodies have already been obtained and the relevance of the presented article has been reduced. Despite the fact that MTX-COVAB’s activity is within the best 10% of neutralizing antibodies against the Wuhan strain, it demonstrates insufficient activity against new coronavirus variants, such as Omicron. In my opinion, the most interesting part of the manuscript is the method of producing antibodies itself, which was developed by the authors. Unfortunately, this part of the paper is described only in the most general terms.

My specific comments:

1.The manuscript is titled “Fast-track discovery of SARS-CoV-2 neutralizing antibodies from human B cells”, but most of the article is devoted to the characterization of the generated antibodies using methods such as SARS-CoV-2 neutralization in virto (section 3.2 and 3.5), in vivo (section 3.3), ADCC, ADCP and CDC (section 3.4), Spike mutant binding (section 3.6). The process of discovery of SARS-CoV-2 neutralizing antibodies is described very briefly and illustrated in Fig. 1, which is very similar to a graphical abstract. Neither the Methods nor the Results describe how the library of paired VH and VL genes was obtained. The lentiviral vectors that were used for transduction, as well as other parameters of this procedure, which are usually given in experimental articles, are not indicated. It is only stated that the DROPZYLLA® platform was used for this. There are no references to this technique where it would be described in more detail. Based on this, the manuscript more deserves the title “Characterization of antibodies generated using the DROPZYLLA® platform”.

2. The Results indicate the number of convalescent COVID-19 patients (n=12), the number of memory B cells (~2 million). To assess the capacity of the resulting library and cellular display, it is advisable to supplement this information with data on the number of viral particles in the library, the number of transduced HEK293 cells (IgG+ and RBD-binding cells), and the number of clones obtained after the first and second round of sorting.

3.Lines 259-260: Sorted clones were once more confirmed by analytical flow cytometry for RBD-binding and absence of huACE2 binding.

It would be desirable to provide a representative analytical flow plots for one or perhaps all four HEK293-IgG hit clones.

4.Fig. 6 shows binding of MTX-COVAB to single point mutations in SARS-CoV-2 spike protein transiently expressed on HEK293T cells. As I understand it, relative binding strength was assessed by the percentage of stained cells. The percentage of Spike+ HEK293 cells may also depend on the transfection efficiency of the particular Spike variant. How did the authors take this circumstance into account?

5.HEK293T-transduced cells were first stained and sorted using double staining with RBD-PE and RBD-APC, and then with RBD-PE and huACE2-AF647. How much time passed between these stainings and sortings? During this time, did mIgG manage to free itself from previous labeling or not?

6.Lines 186-188. “Based on the analysis by FACS for CD56 and CD3 double-positive cells, the isolated NK cell populations used for the assay were >80% pure.

Usually NK cells are defined as CD56+CD3- cells. It is necessary to comment on why the authors used the CD56+CD3+ criterion to determine the purity of the NK population.

7.The phrase“Blood was obtained in accordance with the laws of Switzerland and under ethics approval BASEC-2016-01260 of the Cantonal Ethics Commission of Zurich”, with slight variations, is repeated several times in the text (lines 72, 189, 210, 227). I think it can be given once in section 2.1. Donors & Ethics.

Round 2

Reviewer 1 Report

Comments and Suggestions for Authors

The critiques are addressed.

Reviewer 2 Report

Comments and Suggestions for Authors

Authors have addressed my concerns where possible.

Reviewer 3 Report

Comments and Suggestions for Authors

The authors followed all my comments.